# Cortical microcircuits as
# gated-recurrent neural networks

**Rui Ponte Costa**[*]
Centre for Neural Circuits and Behaviour
Dept. of Physiology, Anatomy and Genetics
University of Oxford, Oxford, UK
`rui.costa@cncb.ox.ac.uk`

**Yannis M. Assael**[*]
Dept. of Computer Science
University of Oxford, Oxford, UK
and DeepMind, London, UK
`yannis.assael@cs.ox.ac.uk`

**Brendan Shillingford**[*]
Dept. of Computer Science
University of Oxford, Oxford, UK
and DeepMind, London, UK
`brendan.shillingford@cs.ox.ac.uk`

**Nando de Freitas**
DeepMind
London, UK
`nandodefreitas@google.com`

**Tim P. Vogels**
Centre for Neural Circuits and Behaviour
Dept. of Physiology, Anatomy and Genetics
University of Oxford, Oxford, UK
`tim.vogels@cncb.ox.ac.uk`

## Abstract

Cortical circuits exhibit intricate recurrent architectures that are remarkably similar across different brain areas. Such stereotyped structure suggests the existence of common computational principles. However, such principles have remained largely elusive. Inspired by gated-memory networks, namely long short-term memory networks (LSTMs), we introduce a recurrent neural network in which information is gated through inhibitory cells that are subtractive (subLSTM). We propose a natural mapping of subLSTMs onto known canonical excitatory-inhibitory cortical microcircuits. Our empirical evaluation across sequential image classification and language modelling tasks shows that subLSTM units can achieve similar performance to LSTM units. These results suggest that cortical circuits can be optimised to solve complex contextual problems and proposes a novel view on their computational function. Overall our work provides a step towards unifying recurrent networks as used in machine learning with their biological counterparts.

## 1   Introduction

Over the last decades neuroscience research has collected enormous amounts of data on the architecture and dynamics of cortical circuits, unveiling complex but stereotypical structures across the neocortex (Markram et al., 2004; Harris and Mrsic-Flogel, 2013; Jiang et al., 2015). One of the most prevalent features of cortical nets is their laminar organisation and their high degree of recurrence, even at the level of local (micro-)circuits (Douglas et al., 1995; Song et al., 2005; Harris and Mrsic-Flogel, 2013; Jiang et al., 2015) (Fig. 1a). Another key feature of cortical circuits is the detailed and tight balance of excitation and inhibition, which has received growing support

---

both at the experimental (Froemke et al., 2007; Xue et al., 2014; Froemke, 2015) and theoretical level (van Vreeswijk and Sompolinsky, 1996; Brunel, 2000; Vogels and Abbott, 2009; Hennequin et al., 2014, 2017). However, the computational processes that are facilitated by these architectures and dynamics are still elusive. There remains a fundamental disconnect between the underlying biophysical networks and the emergence of intelligent and complex behaviours.

Artificial recurrent neural networks (RNNs), on the other hand, are crafted to perform specific computations. In fact, RNNs have recently proven very successful at solving complex tasks such as language modelling, speech recognition, and other perceptual tasks (Graves, 2013; Graves et al., 2013; Sutskever et al., 2014; van den Oord et al., 2016; Assael et al., 2016). In these tasks, the input data contains information across multiple timescales that needs to be filtered and processed according to its relevance. The ongoing presentation of stimuli makes it difficult to learn to separate meaningful stimuli from background noise (Hochreiter et al., 2001; Pascanu et al., 2012). RNNs, and in particular gated-RNNs, can solve this problem by maintaining a representation of relevant input sequences until needed, without interference from new stimuli. In principle, such protected memories conserve past inputs and thus allow back-propagation of errors further backwards in time (Pascanu et al., 2012). Because of their memory properties, one of the first and most successful types of gated-RNNs was named "long short-term memory networks" (LSTMs, Hochreiter and Schmidhuber (1997), Fig. 1c).

Here we note that the architectural features of LSTMs overlap closely with known cortical structures, but with a few important differences with regard to the mechanistic implementation of gates in a cortical network and LSTMs (Fig. 1b). In LSTMs, the gates control the memory cell as a multiplicative factor, but in biological networks, the gates, i.e. inhibitory neurons, act (to a first approximation) subtractively — excitatory and inhibitory (EI) currents cancel each other linearly at the level of the postsynaptic membrane potential (Kandel et al., 2000; Gerstner et al., 2014). Moreover, such a subtractive inhibitory mechanism must be well balanced (i.e. closely match the excitatory input) to act as a gate to the inputs in the 'closed' state, without perturbing activity flow with too much inhibition. Previous models have explored gating in subtractive excitatory and inhibitory balanced networks (Vogels and Abbott, 2009; Kremkow et al., 2010), but without a clear computational role. On the other hand, predictive coding RNNs with EI features have been studied (Bastos et al., 2012; Deneve and Machens, 2016), but without a clear match to state-of-the-art machine learning networks. Regarding previous neuroscientific interpretations of LSTMs, there have been suggestions of LSTMs as models of working memory and different brain areas (e.g. prefrontal cortex, basal ganglia and hippocampus) (O'Reilly and Frank, 2006; Krueger and Dayan, 2009; Cox and Dean, 2014; Marblestone et al., 2016; Hassabis et al., 2017; Bhalla, 2017), but without a clear interpretation of the individual components of LSTMs and a specific mapping to known circuits.

We propose to map the architecture and function of LSTMs directly onto cortical circuits, with gating provided by lateral subtractive inhibition. Our networks have the potential to exhibit the excitation-inhibition balance observed in experiments (Douglas et al., 1989; Bastos et al., 2012; Harris and Mrsic-Flogel, 2013) and yield simpler gradient propagation than multiplicative gating.

We study these dynamics through our empirical evaluation showing that subLSTMs achieve similar performance to LSTMs in the Penn Treebank and Wikitext-2 language modelling tasks, as well as pixelwise sequential MNIST classification. By transferring the functionality of LSTMs into a biologically more plausible network, our work provides testable hypotheses for the most recently emerging, technologically advanced experiments on the functionality of entire cortical microcircuits.

## 2    Biological motivation

The architecture of LSTM units, with their general feedforward structure aided by additional recurrent memory and controlled by lateral gates, is remarkably similar to the columnar architecture of cortical circuits (Fig. 1; see also Fig. S1 for a more detailed neocortical schematic). The central element in LSTMs and similar RNNs is the *memory cell*, which we hypothesise to be implemented by local recurrent networks of pyramidal cells in layer-5. This is in line with previous studies showing a relatively high level of recurrence and non-random connectivity between pyramidal cells in layer-5 (Douglas et al., 1995; Thomson et al., 2002; Song et al., 2005). Furthermore, layer-5 pyramidal networks display rich activity on (relatively) long time scales *in vivo* (Barthó et al., 2009; Sakata and Harris, 2009; Luczak et al., 2015; van Kerkoerle et al., 2017) and in slices (Egorov et al., 2002; Wang et al., 2006), consistent with LSTM-like function. There is strong evidence for persistent

neuronal activity both in higher cortical areas (Goldman-Rakic, 1995) and in sensory areas (Huang et al., 2016; van Kerkoerle et al., 2017; Kornblith et al., 2017). Relatively speaking, sensory areas (e.g. visual cortex) exhibit sorter timescales than higher brain areas (e.g. prefrontal cortex), which we would expect given the different temporal requirements these brain areas have. A similar behaviour is expected in multi-area (or layer) LSTMs. Note that such longer time-scales can also be present in more superficial layers (e.g. layer 2/3) (Goldman-Rakic, 1995; van Kerkoerle et al., 2017), suggesting the possibility of more than one memory cell per cortical microcircuit. Slow memory decay in these networks may be controlled through short- (York and van Rossum, 2009; Costa et al., 2013, 2017a) and long-term synaptic plasticity (Abbott and Nelson, 2000; Senn et al., 2001; Pfister and Gerstner, 2006; Zenke et al., 2015; Costa et al., 2015, 2017a,b) at recurrent excitatory synapses.

The gates that protect a given memory in LSTMs can be mapped onto lateral inhibitory inputs in cortical circuits. We propose that, similar to LSTMs, the input gate is implemented by inhibitory neurons in layer-2/3 (or layer-4; Fig. 1a). Such lateral inhibition is consistent with the canonical view of microcircuits (Douglas et al., 1989; Bastos et al., 2012; Harris and Mrsic-Flogel, 2013) and sparse sensory-evoked responses in layer-2/3 (Sakata and Harris, 2009; Harris and Mrsic-Flogel, 2013). In the brain, this inhibition is believed to originate from (parvalbumin) basket cells, providing a near-exact balanced inhibitory counter signal to a given excitatory feedforward input (Froemke et al., 2007; Xue et al., 2014; Froemke, 2015). Excitatory and inhibitory inputs thus cancel each other and arriving signals are ignored by default. Consequently, any activity within the downstream memory network remains largely unperturbed, unless it is altered through targeted modulation of the inhibitory activity (Harris and Mrsic-Flogel, 2013; Vogels and Abbott, 2009; Letzkus et al., 2015). Similarly, the memory cell itself can only affect the output of the LSTM when its activity is unaccompanied by congruent inhibition (mapped onto layer-5, layer-6 or layer 2/3 in the same microcircuit, which are known to project to higher brain areas (Harris and Mrsic-Flogel, 2013); see Fig. S1), i.e. when lateral inhibition is turned down and the gate is open.

## 2.1 Why subtractive neural integration?

When a presynaptic cell fires, neurotransmitter is released by its synaptic terminals. The neurotransmitter is subsequently bound by postsynaptic receptors where it prompts a structural change of an ion channel to allow the flow of electrically charged ions into or out of the postsynaptic cell. Depending on the receptor type, the ion flux will either increase (depolarise) or decrease (hyperpolarise) the postsynaptic membrane potential. If sufficiently depolarising "excitatory" input is provided, the postsynaptic potential will reach a threshold and fire a stereotyped action potential ("spike", Kandel et al. (2000)). This behaviour can be formalised as a RC–circuit ($R$ = resistance, $C$ = capacitance), which follows Ohm's laws $u = RI$ and yields the standard leaky-integrate-and-fire neuron model (Gerstner and Kistler, 2002), $\tau_m \dot{u} = -u + RI_{\text{exc}} - RI_{\text{inh}}$, where $\tau_m = RC$ is the membrane time constant, and $I_{\text{exc}}$ and $I_{\text{inh}}$ are the excitatory and inhibitory (hyperpolarizing) synaptic input currents, respectively.

Action potentials are initiated in this standard model (Brette and Gerstner, 2005; Gerstner et al., 2014)) when the membrane potential hits a hard threshold $\theta$. They are modelled as a momentary pulse and a subsequent reset to a resting potential. Neuronal excitation and inhibition have opposite effects, such that inhibitory inputs acts linearly and subtractively on the membrane potential.

The leaky-integrate-and-fire model can be approximated at the level of firing rates as rate $\sim \left( \tau_m \ln \frac{R(I_{exc} - I_{inh})}{R(I_{exc} - I_{inh}) - \theta} \right)^{-1}$ (see Fig. 1a for the input-output response; Gerstner and Kistler (2002)), which we used to demonstrate the impact of subtractive gating (Fig. 1b), and contrast it with multiplicative gating (Fig. 1c).

This firing-rate approximation forms the basis for our gated-RNN model which has a similar subtractive behaviour and input-output function (cf. Fig. 1b; bottom). Moreover, the rate formulation also allows a cleaner comparison to LSTM units and the use of existing machine learning optimisation methods.

It could be argued that a different form of inhibition (shunting inhibition), which counteracts excitatory inputs by decreasing the over all membrane resistance, has a characteristic multiplicative gating effect on the membrane potential. However, when analysed at the level of the output firing rate its effect becomes subtractive (Holt and Koch, 1997; Prescott and De Koninck, 2003). This is consistent with

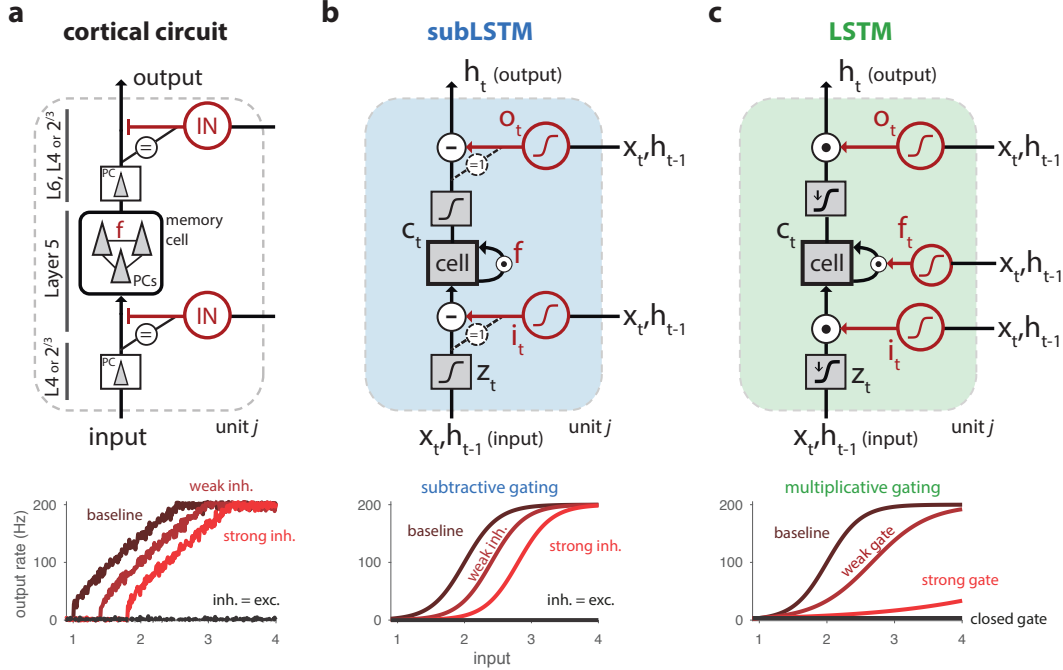

Figure 1: Biological and artificial gated recurrent neural networks. (a) Example unit of a simplified cortical recurrent neural network. Sensory (or downstream) input arrives at pyramidal cells in layer-2/3 (L2/3, or layer-4), which is then fed onto memory cells (recurrently connected pyramidal cells in layer-5). The memory decays with a decay time constant $f$. Input onto layer-5 is balanced out by inhibitory basket cells (BC). The balance is represented by the diagonal 'equal' connection. The output of the memory cell is gated by basket cells at layer-6, 2/3 or 4 within the same area (or at an upstream brain area). (b) Implementation of (a), following a similar notation to LSTM units, but with $\mathbf{i}_t$ and $\mathbf{o}_t$ as the input and output subtractive gates. Dashed connections represent the potential to have a balance between excitatory and inhibitory input (weights are set to 1) (c) LSTM recurrent neural network cell (see main text for details). The plots bellow illustrate the different gating modes: (a) using a simple current-based noisy leaky-integrate-and-fire neuron (capped to 200Hz) with subtractive inhibition; (b) sigmoidal activation functions with subtractive gating; (c) sigmoidal activation functions with multiplicative gating. Output rate represents the number of spikes per second (Hz) as in biological circuits.

our approach in that our model is framed at the firing-rate level (rather than at the level of membrane potentials).

## 3   Subtractive-gated long short-term memory

In an LSTM unit (Hochreiter and Schmidhuber, 1997; Greff et al., 2015) the access to the memory cell $\mathbf{c}_t$ is controlled by an input gate $\mathbf{i}_t$ (see Fig.1c). At the same time a *forget* gate $\mathbf{f}_t$ controls the decay of this memory[1], and the *output* gate $\mathbf{o}_t$ controls whether the content of the memory cell $\mathbf{c}_t$ is transmitted to the rest of the network. A LSTM network consists of many LSTM units, each containing its own memory cell $\mathbf{c}_t$, input $\mathbf{i}_t$, forget $\mathbf{f}_t$ and output $\mathbf{o}_t$ gates. The LSTM state is described as $\mathbf{h}_t = f(\mathbf{x}_t, \mathbf{h}_{t-1}, \mathbf{i}_t, \mathbf{f}_t, \mathbf{o}_t)$ and the unit follows the dynamics given in the middle column below.

|  | **LSTM** | **subLSTM** |
|---|---|---|
| $[\mathbf{f}_t, \mathbf{o}_t, \mathbf{i}_t]^T =$ | $\sigma(W\mathbf{x}_t + R\mathbf{h}_{t-1} + \mathbf{b}),$ | $\sigma(W\mathbf{x}_t + R\mathbf{h}_{t-1} + \mathbf{b}),\,^2$ |
| $\mathbf{z}_t =$ | $\tanh(W\mathbf{x}_t + R\mathbf{h}_{t-1} + \mathbf{b}),$ | $\sigma(W\mathbf{x}_t + R\mathbf{h}_{t-1} + \mathbf{b}),$ |
| $\mathbf{c}_t =$ | $\mathbf{c}_{t-1} \odot \mathbf{f}_t + \mathbf{z}_t \odot \mathbf{i}_t,$ | $\mathbf{c}_{t-1} \odot \mathbf{f}_t + \mathbf{z}_t - \mathbf{i}_t,$ |
| $\mathbf{h}_t =$ | $\tanh(\mathbf{c}_t) \odot \mathbf{o}_t.$ | $\sigma(\mathbf{c}_t) - \mathbf{o}_t.$ |

Here, $\mathbf{c}_t$ is the memory cell (note the multiplicative control of the input gate), $\odot$ denotes element-wise multiplication and $\mathbf{z}_t$ is the new weighted input given with $\mathbf{x}_t$ and $\mathbf{h}_{t-1}$ being the input vector and recurrent input from other LSTM units, respectively. The overall output of the LSTM unit is then computed as $\mathbf{h}_t$. LSTM networks can have multiple layers with millions of parameters (weights and biases), which are typically trained using stochastic gradient descent in a supervised setting. Above, the parameters are $W$, $R$ and $\mathbf{b}$. The multiple gates allow the network to adapt the flow of information depending on the task at hand. In particular, they enable writing to the memory cell (controlled by *input* gate, $\mathbf{i}_t$), adjusting the timescale of the memory (controlled by *forget* gate, $\mathbf{f}_t$) and exposing the memory to the network (controlled by *output* gate, $\mathbf{o}_t$). The combined effect of these gates makes it possible for LSTM units to capture temporal (contextual) dependencies across multiple timescales.

Here, we introduce and study a new RNN unit, subLSTM. SubLSTM units are a mapping of LSTMs onto known canonical excitatory-inhibitory cortical microcircuits (Douglas et al., 1995; Song et al., 2005; Harris and Mrsic-Flogel, 2013). Similarly, subLSTMs are defined as $\mathbf{h}_t = f(\mathbf{x}_t, \mathbf{h}_{t-1}, \mathbf{i}_t, \mathbf{f}_t, \mathbf{o}_t)$ (Fig. 1b), however here the gating is subtractive rather than multiplicative. A subLSTM is defined by a memory cell $\mathbf{c}_t$, the transformed input $\mathbf{z}_t$ and the input gate $\mathbf{i}_t$. In our model we use a simplified notion of balance in the gating $(\theta z_t^j - \theta i_t^j)$ (for the $j$th unit), where $\theta = 1$. [3] For the memory *forgetting* we consider two options: (i) controlled by gates (as in an LSTM unit) as $\mathbf{f}_t = \sigma(W\mathbf{x}_t + R\mathbf{h}_{t-1} + \mathbf{b})$ or (ii) a more biologically plausible learned simple decay $[0, 1]$, referred to in the results as fix-subLSTM. Similarly to its input, subLSTM's output $\mathbf{h}_t$ is also gated through a subtractive *output* gate $\mathbf{o}_t$ (see equations above). We evaluated different activation functions and sigmoidal transformations had the highest performance.

The key differences to other gated-RNNs is in the subtractive inhibitory gating ($\mathbf{i}_t$ and $\mathbf{o}_t$) that has the potential to be balanced with the excitatory input ($\mathbf{z}_t$ and $\mathbf{c}_t$, respectively; Fig. 1b). See below a more detailed comparison of the different gating modes.

## 3.1 Subtractive versus multiplicative gating in RNNs

The key difference between subLSTMs and LSTMs lies in the implementation of the gating mechanism. LSTMs typically use a multiplicative factor to control the amplitude of the input signal. SubLSTMs use a more biologically plausible interaction of excitation and inhibition. An important consequence of subtractive gating is the potential for an improved gradient flow backwards towards the input layers. To illustrate this we can compare the gradients for the subLSTMs and LSTMs in a simple example.

First, we review the derivatives of the loss with respect to the various components of the subLSTM, using notation based on (Greff et al., 2015). In this notation, $\delta \mathbf{a}$ represents the derivative of the loss

with respect to $\mathbf{a}$, and $\Delta_t \stackrel{\text{def}}{=} \frac{d\text{loss}}{d\mathbf{h}_t}$, the error from the layer above. Then by chain rule we have:

$$\delta\mathbf{h}_t = \Delta_t$$
$$\delta\overline{\mathbf{o}}_t = -\delta\mathbf{h}_t \odot \sigma'(\overline{\mathbf{o}}_t)$$
$$\delta\mathbf{c}_t = \delta\mathbf{h}_t \odot \sigma'(\mathbf{c}_t) + \delta\mathbf{c}_{t+1} \odot \mathbf{f}_{t+1}$$
$$\delta\overline{\mathbf{f}}_t = \delta\mathbf{c}_t \odot \mathbf{c}_{t-1} \odot \sigma'(\overline{\mathbf{f}}_t)$$
$$\delta\overline{\mathbf{i}}_t = -\delta\mathbf{c}_t \odot \sigma'(\overline{\mathbf{i}}_t)$$
$$\delta\overline{\mathbf{z}}_t = \delta\mathbf{c}_t \odot \sigma'(\overline{\mathbf{z}}_t)$$

For comparison, the corresponding derivatives for an LSTM unit are given by:

$$\delta\mathbf{h}_t = \Delta_t$$
$$\delta\overline{\mathbf{o}}_t = \mathbf{h}_t \odot \tanh(\mathbf{c}_t) \odot \sigma'(\overline{\mathbf{o}}_t)$$
$$\delta\mathbf{c}_t = \mathbf{h}_t \odot \mathbf{o}_t \odot \tanh'(\mathbf{c}_t) + \delta\mathbf{c}_{t+1} \odot \mathbf{f}_{t+1}$$
$$\delta\overline{\mathbf{f}}_t = \delta\mathbf{c}_t \odot \mathbf{c}_{t-1} \odot \sigma'(\overline{\mathbf{f}}_t)$$
$$\delta\overline{\mathbf{i}}_t = \delta\mathbf{c}_t \odot \mathbf{z}_t \odot \sigma'(\overline{\mathbf{i}}_t)$$
$$\delta\overline{\mathbf{z}}_t = \delta\mathbf{c}_t \odot \mathbf{i}_t \odot \tanh'(\overline{\mathbf{z}}_t)$$

where $\sigma(\cdot)$ is the sigmoid activation function and the overlined variables $\overline{\mathbf{c}}_t, \overline{\mathbf{f}}_t$, etc. are the pre-activation values of a gate or input transformation (e.g. $\overline{\mathbf{o}}_t = W_o\mathbf{x}_t + R_o\mathbf{h}_{t-1} + b_o$ for the output gate of a subLSTM). Note that compared to the those of an LSTM, subLSTMs provide a simpler gradient with fewer multiplicative factors.

Now, the LSTMs weights $W_z$ of the input transformation $\mathbf{z}$ are updated according to

$$\delta W_z = \sum_{t=0}^{T} \sum_{t'=t}^{T} \Delta_{t'} \frac{\partial\mathbf{h}_{t'}}{\partial\mathbf{c}_{t'}} \cdots \frac{\partial\mathbf{c}_t}{\partial\mathbf{z}_t} \frac{\partial\mathbf{z}_t}{\partial W_z}, \tag{1}$$

where $T$ is the total number of temporal steps and the ellipsis abbreviates the recurrent gradient paths through time, containing the path backwards through time via $\mathbf{h}_s$ and $\mathbf{c}_s$ for $t \leq s \leq t'$. For simplicity of analysis, we ignore these recurrent connections as they are the same in LSTM and subLSTM, and only consider the depth-wise path through the network; we call this $t$th timestep depth-only contribution to the derivative $(\delta W_z)_t$. For an LSTM, by this slight abuse of notation, we have

$$(\delta W_z)_t = \Delta_t \frac{\partial\mathbf{h}_t}{\partial\mathbf{c}_t} \frac{\partial\mathbf{c}_t}{\partial\mathbf{z}_t} \frac{\partial\mathbf{z}_t}{\partial\overline{\mathbf{z}}_t} \frac{\partial\overline{\mathbf{z}}_t}{\partial W_z}$$
$$= \left( \Delta_t \odot \underbrace{\mathbf{o}_t}_{\text{output gate}} \odot \tanh'(\overline{\mathbf{c}}_t) \odot \underbrace{\mathbf{i}_t}_{\text{input gate}} \odot \tanh'(\overline{\mathbf{z}}_t) \right)\mathbf{x}_t^\top, \tag{2}$$

where $\tanh'(\cdot)$ is the derivative of $\tanh$. Notice that when either of the input or output gates are set to zero (closed), the corresponding contributions to the gradient are zero. For a network with subtractive gating, the depth-only derivative contribution becomes

$$(\delta W_z)_t = \left( \Delta_t \odot \sigma'(\overline{\mathbf{c}}_t) \odot \sigma'(\overline{\mathbf{z}}_t) \right)\mathbf{x}_t^\top, \tag{3}$$

where $\sigma'(\cdot)$ is the sigmoid derivative. In this case, the input and output gates, $\mathbf{o}_t$ and $\mathbf{i}_t$, are not present. As a result, the subtractive gates in subLSTMs do not (directly) impair error propagation.

## 4 Results

The aims of our work were two-fold. First, inspired by cortical circuits we aimed to propose a biological plausible implementation of an LSTM unit, which would allow us to better understand cortical architectures and their dynamics. To compare the performance of subLSTM units to LSTMs, we first compared the learning dynamics for subtractive and multiplicative networks mathematically. In a second step, we empirically compared subLSTM and fix-subLSTM with LSTM networks in

two tasks: sequential MNIST classification and word-level language modelling on Penn Treebank (Marcus et al., 1993) and Wikitext-2 (Merity et al., 2016). The network weights are initialised with Glorot initialisation (Glorot and Bengio, 2010), and LSTM units have an initial forget gate bias of 1. We selected the number of units for fix-subLSTM such that the number of parameters is held constant across experiments to facilitate fair comparison with LSTMs and subLSTMs.

## 4.1  Sequential MNIST

In the "sequential" MNIST digit classification task, each digit image from the MNIST dataset is presented to the RNN as a sequence of pixels (Le et al. (2015); Fig. 2a) We decompose the MNIST images of $28 \times 28$ pixels into sequences of 784 steps. The network was optimised using RMSProp with momentum (Tieleman and Hinton, 2012), a learning rate of $10^{-4}$, one hidden layer and 100 hidden units. Our results show that subLSTMs achieves similar results to LSTMs (Fig. 2b). Our results are comparable to previous results using the same task (Le et al., 2015) and RNNs.

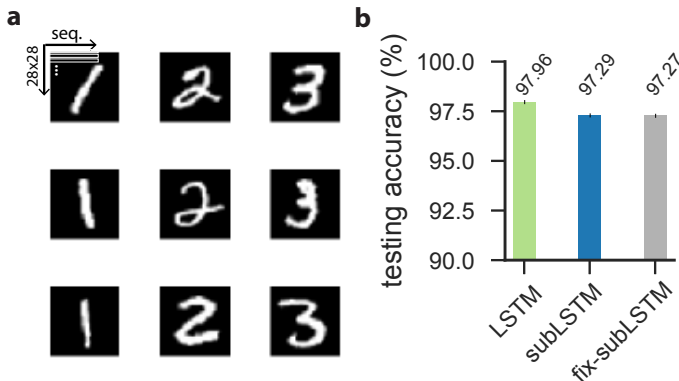

Figure 2: Comparison of LSTM and subLSTM networks for sequential pixel-by-pixel MNIST, using 100 hidden units. (**a**) Samples from MNIST dataset. We converted each matrix of $28 \times 28$ pixels into a temporal sequence of 784 timesteps. (**b**) Classification accuracy on the test set. fix-subLSTM has a fixed but learned forget gate.

## 4.2  Language modelling

Language modelling represents a more challenging task for RNNs, with both short and long-term dependencies. RNN language models (RNN LMs) models the probability of text by autoregressively predicting a sequence of words. Each timestep is trained to predict the following word; in other words, we model the word sequence as a product of conditional multinoulli distributions. We evaluate the RNN LMs by measuring their perplexity, defined for a sequence of $n$ words as

$$\text{perplexity} = P(w_1, \ldots, w_n)^{-1/n}. \tag{4}$$

We first used the Penn Treebank (PTB) dataset to train our model on word-level language modelling (929k training, 73k validation and 82k test words; with a vocabulary of 10k words).

All RNNs tested have 2 hidden layers; backpropagation is truncated to 35 steps, and a batch size of 20. To optimise the networks we used RMSProp with momentum. We also performed a hyperparameter search on the validation set over input, output, and update dropout rates, the learning rate, and weight decay. The hyperparameter search was done with Google Vizier, which performs black-box optimisation using Gaussian process bandits and transfer learning. Tables 2 and 3 show the resulting hyperparameters. Table 1 reports perplexity on the test set (Golovin et al., 2017). To understand how subLSTMs scale with network size we varied the number of hidden units between 10, 100, 200 and 650.

We also tested the Wikitext-2 language modelling dataset based on Wikipedia articles. This dataset is twice as large as the PTB dataset (2000k training, 217k validation and 245k test words) and also

features a larger vocabulary (33k words). Therefore, it is well suited to evaluate model performance on longer term dependencies and reduces the likelihood of overfitting.

On both datasets, our results show that subLSTMs achieve perplexity similar to LSTMs (Table 1a and 1b). Interestingly, the more biological plausible version of subLSTM (with a simple decay as forget gates) achieves performance similar to or better than subLSTMs.

| (a) Penn Treebank (PTB) test perplexity | | | | (b) Wikitext-2 test perplexity | | | |
|---|---|---|---|---|---|---|---|
| size | subLSTM | fix-subLSTM | LSTM | size | subLSTM | fix-subLSTM | LSTM |
| 10 | 222.80 | 213.86 | 215.93 | 10 | 268.33 | 259.89 | 271.44 |
| 100 | 91.46 | 91.84 | 88.39 | 100 | 103.36 | 105.06 | 102.77 |
| 200 | 79.59 | 81.97 | 74.60 | 200 | 89.00 | 94.33 | 86.15 |
| 650 | 76.17 | 70.58 | 64.34 | 650 | 78.92 | 79.49 | 74.27 |

Table 1: Language modelling (word-level) test set perplexities on (a) Penn Treebank and (b) Wikitext-2. The models have two layers and fix-subLSTM uses a fixed but learned forget gate $f = [0, 1]$ for each unit. The number of units for fix-subLSTM was chosen such that the number of parameters were the same as those of (sub)LSTM to facilitate fair comparison. Size indicates the number of units.

The number of hidden units for fix-subLSTM were selected such that the number of parameters were the same as LSTM and subLSTM, facilitating fair comparison.

## 5   Conclusions & future work

Cortical microcircuits exhibit complex and stereotypical network architectures that support rich dynamics, but their computational power and dynamics have yet to be properly understood. It is known that excitatory and inhibitory neuron types interact closely to process sensory information with great accuracy, but making sense of these interactions is beyond the scope of most contemporary experimental approaches.

LSTMs, on the other hand, are a well-understood and powerful tool for contextual tasks, and their structure maps intriguingly well onto the stereotyped connectivity of cortical circuits. Here, we analysed if biologically constrained LSTMs (i.e. subLSTMs) could perform similarly well, and indeed,

| Model | hidden units | input dropout | output dropout | update dropout | learning rate | weight decay |
|---|---|---|---|---|---|---|
| LSTM | 10 | 0.026 | 0.047 | 0.002 | 0.01186 | 0.000020 |
| subLSTM | 10 | 0.012 | 0.045 | 0.438 | 0.01666 | 0.000009 |
| fix-subLSTM | 11 | 0.009 | 0.043 | 0 | 0.01006 | 0.000029 |
| LSTM | 100 | 0.099 | 0.074 | 0.015 | 0.00906 | 0.000532 |
| subLSTM | 100 | 0.392 | 0.051 | 0.246 | 0.01186 | 0.000157 |
| fix-subLSTM | 115 | 0.194 | 0.148 | 0.042 | 0.00400 | 0.000218 |
| LSTM | 200 | 0.473 | 0.345 | 0.013 | 0.00496 | 0.000191 |
| subLSTM | 200 | 0.337 | 0.373 | 0.439 | 0.01534 | 0.000076 |
| fix-subLSTM | 230 | 0.394 | 0.472 | 0.161 | 0.00382 | 0.000066 |
| LSTM | 650 | 0.607 | 0.630 | 0.083 | 0.00568 | 0.000145 |
| subLSTM | 650 | 0.562 | 0.515 | 0.794 | 0.00301 | 0.000227 |
| fix-subLSTM | 750 | 0.662 | 0.730 | 0.530 | 0.00347 | 0.000136 |

Table 2: Penn Treebank hyperparameters.

| Model | hidden units | input dropout | output dropout | update dropout | learning rate | weight decay |
|---|---|---|---|---|---|---|
| LSTM | 10 | 0.015 | 0.039 | 0 | 0.01235 | 0 |
| subLSTM | 10 | 0.002 | 0.030 | 0.390 | 0.00859 | 0.000013 |
| fix-subLSTM | 11 | 0.033 | 0.070 | 0.013 | 0.00875 | 0 |
| LSTM | 100 | 0.198 | 0.154 | 0.002 | 0.01162 | 0.000123 |
| subLSTM | 100 | 0.172 | 0.150 | 0.009 | 0.00635 | 0.000177 |
| fix-subLSTM | 115 | 0.130 | 0.187 | 0 | 0.00541 | 0.000172 |
| LSTM | 200 | 0.379 | 0.351 | 0 | 0.00734 | 0.000076 |
| subLSTM | 200 | 0.342 | 0.269 | 0.018 | 0.00722 | 0.000111 |
| fix-subLSTM | 230 | 0.256 | 0.273 | 0 | 0.00533 | 0.000160 |
| LSTM | 650 | 0.572 | 0.566 | 0.071 | 0.00354 | 0.000112 |
| subLSTM | 650 | 0.633 | 0.567 | 0.257 | 0.00300 | 0.000142 |
| fix-subLSTM | 750 | 0.656 | 0.590 | 0.711 | 0.00321 | 0.000122 |

Table 3: Wikitext-2 hyperparameters.

such subtractively gated excitation-inhibition recurrent neural networks show promise compared against LSTMs [4] on benchmarks such as sequence classification and word-level language modelling.

While it is notable that subLSTMs could not outperform their traditional counterpart (yet), we hope that our work will serve as a platform to discuss and develop ideas of cortical function and to establish links to relevant experimental work on the role of excitatory and inhibitory neurons in contextual learning (Froemke et al., 2007; Froemke, 2015; Poort et al., 2015; Pakan et al., 2016; Kuchibhotla et al., 2017). In future work, it will be interesting to study how additional biological detail may affect performance. Next steps should aim to include Dale's principle (i.e. that a given neuron can only make either excitatory or inhibitory connections, Strata and Harvey (1999)), and naturally focus on the perplexing diversity of inhibitory cell types (Markram et al., 2004) and behaviour, such as shunting inhibition and mixed subtractive and divisive control (Doiron et al., 2001; Mejias et al., 2013; El Boustani and Sur, 2014; Seybold et al., 2015).

Overall, given the success of multiplicative gated LSTMs, it will be most insightful to understand if some of the biological tricks of cortical networks may give LSTMs a further performance boost.

### Acknowledgements

We would like to thank Everton Agnes, Çağlar Gülçehre, Gabor Melis and Jake Stroud for helpful comments and discussion. R.P.C. and T.P.V. were supported by a Sir Henry Dale Fellowship by the Wellcome Trust and the Royal Society (WT 100000). Y.M.A. was supported by the EPSRC and the Research Council UK (RCUK). B.S. was supported by the Clarendon Fund.

## Footnotes

[1]Note that this leak is controlled by the input and recurrent units, which may be biologically unrealistic.

[2]Note that we consider two versions of subLSTMs: one with a forget gate as in LSTMs (subLSTM) and another with a simple memory decay (i.e. a scalar [0,1] that defines the memory timeconstant, fix-subLSTM).

[3]These weights could also be optimised, but for this model we decided to keep the number of parameters to a minimum for simplicity and ease of comparison with LSTMs.

[4]Although here we have focus on a comparison with LSTMs, similar points would also apply to other gated-RNNs, such as Gated Recurrent Units (Chung et al., 2014).

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
