[Supplementary Material]

## Supplemental Material: Cortical microcircuits as gated-recurrent neural networks

## A    Detailed view of cortical microcircuit mapping

In Fig. S1 we use a standard representation of the six neocortical layers to highlight the features that are akin to properties of gated-RNNs, namely LSTMs.

Figure S1: Biological properties of the neocortex that resemble LSTM features (cf. Figure 1). Here we represent the different components as a sequence of steps starting at the input level, and highlight the features that are analogous to LSTMs. (1) The sensory input arrives (from thalamus) at layer-4 stellate excitatory cells (faded out as this cell is not explicitly considered in the model) and is then propagated to layer 2/3 pyramidal cells (i.e. the input neuron $z$ in LSTMs); (2) Output of layer 2/3 pyramidal cells projects to layer-5 pyramidal cells which may form recurrent units (the memory cell); At this stage local interneurons (input gates $i$ in LSTMs) gate the flow of information onto the memory cell; (3) This memory cell can retain a given input for sometime (the timescale of this memory will depend on $f$, see subLSTM description in the main text). (4) Finally the memory is sent to other units in the network (black dashed arrow pointing upwards, through layer 2/3) or to another set of units in higher cortical areas (black dashed arrow pointing rightwards, through layer 5 or layer 6). This output of the memory cell is controlled by another gate (local inhibitory cell, representing the output gate $o$). The balance between excitation-inhibition observed across the cortex is represented by the 'equal' connection, which represents equal and fixed weights (i.e. =1, see main text).