[Reviews · NeurIPS 2017]

Reviewer 1



A new recurrent neural network model is presented. It has similar functional features as LSTMs but additive gating instead of multiplicative gating for the input and output gates. With this additive gating mechanism there are some striking similarities to cortical circuits. In my opinion, this papers could be really interesting for both computational neuroscientists and machine learners. For small network sizes the proposed model performs as good or better than simple LSTMs on some non-trivial tasks. It is, however, somewhat disappointing that For the language modelling task it seems, however, that the multiplicative forget gates are still needed for good performance in larger networks. Further comments and questions: 1. I guess the sigma-subRNN have also controlled forget gates (option i on line 104). If yes, it would be fair to highlight this fact more clearly in the results/discussion section. It is somewhat disappointing that this not so biologically plausible feature seems to be needed to achieve good performances on the language modelling tasks. 2. Would it be possible to implement the controlled forget gate also with a subtractive gate. Maybe something like c_t = relu(c_{t-1} - f_t) + (z_t - i_t), which could be interpreted as a strong inhibition (f_t) reseting the reverberating loop before new input (z_t - i_t) arrives. But maybe the sign of c_t is important.... 3. line 111: Why is it important to have rectified linear transfer functions? Doesn't sigma-subRNN perform better in some cases? From the perspective of biological plausibility the transfer function should have the range [0, f_max] where f_max is the maximal firing rate of a neuron (see e.g. the transfer function of a LIF neuron with absolute refractoriness). 4. line 60 "inhibitory neurons act subtractively": I think one should acknowledge that the picture is not so simple in biological systems (e.g. Chance et al. 2002 https://doi.org/10.1016/S0896-6273(02)00820-6 or Müllner et al. 2015 https://doi.org/10.1016/j.neuron.2015.07.003). 5. In my opinion it is not needed to explicitely state the well known Gershgorin circle theorem in the main text. 6. Fig 3f: BGRNN? Should this be subRNN? I read through the author’s rebuttal and my colleagues’ review, and decided to not change my assessment of the paper.

Reviewer 2



The authors introduce a new variant of a gated unit for recurrent neural networks. The performance of the RNN with this unit is found to be similar or superior to a standard LSTM. An analysis indicates possible reasons for this performance. In addition, the authors claim that the new unit is similar to the cortical circuit in biological brains. There are two separate strands to this work – machine learning and neuroscience. The neuroscience part is rather weak (detailed below), and I think the work should only be judged according to machine learning merit. It is true that subtraction is in general considered more biologically plausible than multiplication. In this respect the current work moves us a step towards biological realism. The overall biological motivation, however, is not very strong. For instance, the authors claim that “excitatory and inhibitory currents directly cancel each other linearly at the level of the postsynaptic membrane potential”, thereby ignoring conductances. In particular, the concept of shunting inhibition is an example of a case where inhibition does not simply linearly combine with excitation. The interpretation of balanced excitation and inhibition, as motivating the term (z-i) in the model is also not clear. The fact the both ‘z’ and ‘i’ have the same coefficient is not enough for balance. Their magnitude should be the same. The authors do not claim or show that the difference (z-i) is close to zero in all network neurons. A further point that is hard to understand is the way the forget gate is used. It seems that this part remains multiplicative (and a footnote mentions this is not very plausible). At another section, however, the authors note that they use “a more biologically plausible simple decay [0, 1]”, but it’s hard to understand what is exactly meant here. Regarding the machine learning value, the results are not dramatically different from LSTMs, and the comparison less rigorous than e.g. [1]. [1] Greff, Klaus, et al. "LSTM: A search space odyssey." IEEE transactions on neural networks and learning systems (2017).

Reviewer 3



The paper describes how networks of gating balanced inhibition and excitation can be constructed that are capable of learning similar to LSTM. I found the idea proposed in the paper very interesting. The authors however argue bit strong that the cortical organization maps to RNNs: others have argued other mappings (ie predictive coding [1], or [2]). While the mapping may fit, I would argue that that is not sufficient as "proof". I found the paragraph in 4.1 a bit confusing, it seems to suggest the gradient vanishes for LSTMs, whereas this really only applies to the current input and current output. on l131, it is stated that the error-flow to the input layer in an LSTM unit vanishes when one of the gates is 'closed' This is technically true indeed, however, since the gates are fed by sigmoid activations, gates are rarely truly 0. Figure 2 is very hard to understand because the colors for LSTM vs sub gating come out too similar, same for Fig 3 f and Fig 5a The temporal MNIST task could benefit from a bit more detail to explain what is going on. How does the presented LSTM performance compare to the original performance reported by Hochreiter & Schmidhuber? [1] Bastos, Andre M., et al. "Canonical microcircuits for predictive coding." Neuron 76.4 (2012): 695-711. [2] George, Dileep, and Jeff Hawkins. "Towards a mathematical theory of cortical micro-circuits." PLoS computational biology 5.10 (2009): e1000532.